# Effect of high-resolution geostationary satellite imageries on predictability of tropical thunderstorms over Southeast Asia

Kwonmin Lee[1], Hye-Sil Kim[2] and Yong-Sang Choi[1]*

[1]Department of Climate and Energy Systems Engineering, Ewha Womans University, Seoul, South Korea
[2]Department of Atmospheric Science and Engineering, Ewha Womans University, Seoul, South Korea

*Correspondence to*: Prof. Yong-Sang Choi (ysc@ewha.ac.kr)

**Abstract.**

Tropical thunderstorms cause significant damage to property and lives, and a strong research interest exists in the advance and improvement of the thunderstorm predictability by satellite observations. Using high-resolution (2 km and 10 minutes) imageries from the geostationary satellite (Himawari-8), recently launched over Southeast Asia, we examined the earliest possible time for prediction of thunderstorms as compared to the potential of low-resolution (4 km and 30 minutes) imageries of the former satellite. We compared the lead times of high- and low-resolution imageries of 60 tropical thunderstorms that occurred in August 2017. These thunderstorms were identified by the decreasing trend in the 10.45-µm brightness temperature (BT11) by over 5 K per 10 minutes for the high-resolution imagery or 15 K per 30 minutes for the low-resolution imagery. The lead time was then calculated over the time from the initial state to the mature state of the thunderstorm, based on the time series of a minimum BT11 of thunderstorm pixels. The lead time was found to be 90−180 minutes for the high-resolution imagery, whereas it was only 60 minutes (if detectable) for the low-resolution imagery. These results indicate that high-resolution imagery is essential for substantial disaster mitigation owing to its ability to raise an alarm more than two hours ahead of the mature state of a tropical thunderstorm.

# 1 Introduction

Climate change adaptation and disaster risk management integration have become increasingly important issues since unpredictable natural hazards have started to appear more frequently and intensively in the recent warmer climate (Shaw et al., 2010) Impacts from recent climate-related extremes, such as heat waves, droughts, floods, cyclones, and wildfires, reveal the significant vulnerability and exposure of some ecosystems and many human systems to the current climate variability (Pachauri and Meyer, 2014). These severe events cause extensive economic losses, environmental degradation, and subsequent damage to human life.

Multiple hazards can be involved simultaneously or in quick succession in an extreme weather event. For example, a tropical storm can lead to flooding, storm surge, coastal inundation and mudslides with high winds and heavy rain (Nastos et al., 2016). Thus, it is important to quickly detect the early clouds before they have reached their mature state to reduce disaster damage. This early detection and prediction are based on the high possibility that early clouds can be accompanied by heavy rain or lightning in their mature state (Houze Jr and Betts, 1981). Aimed at disaster risk reduction, the European Operational Program for Exchange of Weather Radar Information (OPERA) continuously provides precipitation data of higher spatial resolution over a large area (Huuskonen et al., 2014). The Next Generation Weather Radar (NEXRAD) system (Klazura and Imy, 1993) has been employed for this purpose in the United States.

However, it is challenging to predict a tropical convective system, because numerical weather prediction models generally have coarser spatiotemporal resolution than a deep convective clouds area with short life spans (Avotniece et al., 2017). In Southeast Asia, not only is the prediction model imperfect, but the observational data that support it are also insufficient. To make matters worse, unlike in the middle latitudes, the tropical atmosphere is conditionally unstable, hindering the predictive accuracy of tropical thunderstorms by such models. Hence, alarms for the hazards in the tropics are generally managed by the nowcasting system by real-time observations from radar and meteorological satellites.

The usefulness of the observations by geostationary satellites has been particularly emphasized for convective clouds, considering their extensive spatial coverage (Escrig et al., 2013). Satellites can observe clouds regardless of their location (over land or ocean) and can also monitor the spans of clouds along their tracks (de Coning et al., 2015). Furthermore, geostationary satellite imagers have greatly advanced in recent years. For example, high-resolution interval imagery of 2 km resolution and 10 minutes by the Himawari-8 satellite operated by the Japan Meteorological Agency has been primarily available from the geostationary orbit since 2015 (Bessho et al., 2016). Similar resolution imageries can be obtained by several geostationary satellites: Geostationary Operational Environmental Satellites (GOES) (Menzel and Purdom, 1994), Fengyun-4 (FY-4) (Yang et al., 2017), and GEO-KOMPSAT-2A (GK2A) (Choi and Ho, 2015). These high-resolution imageries have high potential to improve thunderstorm monitoring. Moreover, these satellites are not susceptible to hazards, and continuously provide enhanced thunderstorm information, much wider, more specific, and more accurate that that obtained by ground measurements.

As the number of recent meteorological geostationary satellites loading the enhanced imager increases, people expect to receive higher quality weather information that was unavailable in the past. However, there has been a lack of prior attempts to quantify the predictability of thunderstorms using geostationary infrared imagers. In particular, it has been uncertain how much earlier thunderstorms can be predicted using the improved imagery. Therefore, in this study, we focused on the dependence of tropical thunderstorm predictability on spatiotemporal imagery resolutions. We investigated the predictability of the initial state of thunderstorms over Southeast Asia using the Himawari-8 satellite.

## 2 Data and Method

The region examined in this study is from 10°N to 20°N and from 100°E to 120°E and is closely monitored by the Mekong River Commission. The Mekong River Commission is the only inter-governmental organization interacting directly with the governments of Cambodia, Lao PDR, Thailand, and Viet Nam to jointly manage the shared water resources and the sustainable development of the Mekong River (Jacobs, 2002). Unfortunately, this area is known as a vulnerable-disaster region because of its high risk of extreme weather. The global impacts of climate change have contributed to changes in the weather patterns that are felt across the Mekong River Commission region. The warmer atmosphere has the potential to contain more moisture, which increases the possibilities for invigorating thunderstorms if other conditions are equal. The temperature increase associated with global climate change was generally assumed to lead to increased thunderstorm intensity and associated heavy precipitation events (Schefczyk et al., 2015).

Disasters occurring after tropical thunderstorms are distinctly common during the wet season. The Himawari-8 AHI RGB image of the study area, taken at 05:50 UTC on 19 August 2015, can be seen in Figure 1. Several convective clouds (white color) are shown in the southern part of the area. Normally, in the diurnal cycle of the tropics, oceanic deep convection generally tends to reach its maximum in the morning and continental convection peaks in the evening, although there are interesting regional variations (Yang and Slingo, 2001). Therefore, we conducted observations within the intervals 03:00–06:50 UTC (daytime) and 21:00–24:50 UTC (nighttime), when the potential frequently thunderstorm occurrence is higher due to the diurnal cycle specific for the tropics. Table 1 lists the number of selected convective clouds that are developing within two hours. Sixty clouds, except for those the already developed, were only selected the ones that whose cloudy pixels first began to detect at 03:00–06:50 UTC (daytime) and 21:00–24:50 UTC (nighttime) during in July and August 2017.

Himawari-8 is one of the several launched geostationary meteorological satellites capable of observing Southeast Asia. Himawari-8 has 16 spectral bands, 11 more than the previous satellite, MTSAT-1R/2, as presented in Table 2. Specifically, the spatial resolution is doubled and the time interval is tripled as compared to those of the former. Himawari-8 can scan five areas, each of which with a different time cycle. The area of Japan and the target area were observed every 2.5 minutes, and the landmark area was observed every 0.5 minutes (Table 2). However, in the region of interest of this study, convective cloud observations were possible every 10 minutes based on the full disk (JMA/MSC: Himawari-8/9 Imagery (AHI), 2018).

Among the 16 existing bands, the brightness temperature at 10.45 µm (BT11) was used for monitoring the vertical growth of clouds. The wavelength of 10.45 µm was less sensitive to ozone or water vapor in the atmosphere than any other of the infrared window bands (Schmit et al., 2005). Thus, BT11 for cumulus/convective clouds was closely related to the cloud top temperature. The colder cloud top and larger cloud thickness as the clouds developed vertically effectively reduced BT11. Of note, in this study, we used only the infrared band at 10.45 µm, which is the channel common to both Himawari-8 and MTSAT-1R/2.

To perform this study, we created virtual data whose resolution was similar to those of MTSAT 1R/2 (Table 2). Specifically, four pixels of 2 km were converted into one pixel of 4 km, and the time interval was increased from 10 minutes to 30 minutes. In other words, it was calculated as the average of four 2-km pixels in the process of observing clouds every 30 minutes. The number of detected cloudy pixels by resolution is illustrated in Figure 2. A tropical thunderstorm was found to be located in the area of 12 × 12-km pixels. The dark grey indicates the detected cloudy pixel, and the light grey indicates the clear-sky pixel. Using the 4-km resolution imagery only 2 cloudy pixels were detected in the middle area with the 4-km resolution imagery; in contrast, 18 cloudy pixels can be detected with the 2-km resolution imagery. It is noteworthy that the high-resolution imagery was able to detect cloudy pixels located at a curved boundary. However, the low-resolution imagery tended to simplify the change rate of minimum BT11, and the detection of cloudy pixels at a curved boundary was somewhat hard (Walker et al., 2012). Hereafter, the virtual MTSAT is called the low-resolution (4 km and 30 minutes) imagery and the Himawari-8 is called the high-resolution (2 km and 10 minutes) imagery so as to facilitate the intuitive understanding of the spatiotemporal resolution difference. Our final study aim was to quantitatively compare their effectiveness in the advanced predictability of tropical thunderstorms through the imageries of geostationary satellite.

## 3 Determination of thunderstorm pixels and the lead time

Sixty thunderstorms were subjectively selected based on the RGB images over Southeast Asia. The size of the selected thunderstorms was determined to be less than 120 km because such convective scales typically accompany precipitation (Houze Jr, 2004). We set the rectangular target boundaries depending on the thunderstorm size. In the target boundary, the BT11 values were monitored to determine the thunderstorm pixels and phases (initial/mature states) for the whole life cycle of thunderstorms. Since temporal changes in BT11 inform vertical drift velocity, the current status of clouds can be a key to diagnose the probability of imminent heavy rains/lightning soon (Vila et al., 2008).

The thunderstorm pixels in the target boundaries were identified by the decreasing trend in BT11 by over 5 K per 10 minutes for the high-resolution imagery or 15 K per 30 minutes for the low-resolution imagery. During the observation time, the initial state was defined as the moment when the thunderstorm pixels were firstly detected in the target boundaries. The mature state was defined as the latest moment when the minimum BT11 among thunderstorm pixels decreases gradually below 230 K. The minimum BT11 near 230 K meant that the ice cloud effective radius in the thunderstorms is maximized (Kahn et al., 2018). The large ice cloud effective radius is closely related to rain formation because the cloud drop size has an impact on the cloud growth rate (Wang, 2013).

The lead time was defined as the time between the initial state and the mature state. Figure 3 clearly shows the method used for the calculation of the lead time from the initial/mature state of a thunderstorm. The solid on Figure 3 represents an example of the temporal changes in minimum BT11 among thunderstorm pixels (10 August 2017, 03:10–05:50 UTC). The high-resolution imagery is depicted as the circle and the low-resolution imagery as the triangle. The negative sign of time indicates the time ahead of the mature state of a tropical thunderstorm. Consequentially, the lead time of the targeted thunderstorm for the low-resolution imagery was 30 minutes, and that for the high-resolution imagery was 160 minutes.

The most interesting point of Figure 3 is the pattern of temporal changes in minimum BT11 among the thunderstorm pixels for high-resolution imagery. One can expect that BT11 of a thunderstorm might gradually decrease, but the BT11 of the targeted thunderstorm firstly decrease from the initial state to −70 minutes and increase slightly from −70 minutes to −30 minutes. We speculated that the decline of BT11 was related to the vertical growth of the cloud. This is commonly observed in the life cycle of tropical thunderstorms. It is notable that BT11 for the low-resolution imagery was too simple to monitor this status of the clouds in detail (the triangle in Figure 3).

Table 3 lists the lead time of each imagery type. In 60 cloud samples, No. 1−20 occurred in the daytime of July 2017, No. 21−30 in the nighttime of July 2017, No. 31−50 in the daytime of August 2017, and No. 51−60 in the nighttime of August 2017. In the cases of the high-resolution imagery, the lead time ranged from 90 minutes to 180 minutes. In contrast, the low-resolution imagery began to detect cloud pixels only up to 0−30 minutes in advance. Here, a zero lead time indicates that the low-resolution imagery failed to detect the cloud pixel while the cloud observed with the high-resolution imagery reached the mature state. In this study, the high-resolution imagery had high predictability of thunderstorms, which was up to 180 minutes. The low-resolution imagery, on the other hand, had low predictability of thunderstorms (up to 60 minutes).

Figure 4 depicts the spatial distribution of BT11 used to determine the cause of the difference in the lead time between the different resolutions. Figure 4a and 4b are the result of the low-resolution imagery and that of the high-resolution imagery (10 August 2017, 03:00–05:50 UTC). The amount of change of BT11 within the observation time interval is closely related to determining the initial state. In Figure 4a, pixels with a variation of 15 K or more are recognized as thunderstorm every 30 minutes. In contrast, in Figure 4b, pixels with a variation of 5 K or more are recognized as thunderstorm every 10 minutes. This can cause difficulty to detect the initial state using low-resolution imagery because of the BT11 variation which is proportional to the observation time interval. Also, Figure 4a shows that the boundaries of clouds are not clear from 03:00 to 04:00 UTC and there seems to be a limitation to track continuously developing storms from 04:00 to 05:00 UTC because the low-resolution imagery could not detect cloudy pixels whose scale was between 2 km and 4 km. Figure 4b shows the BT11 at the boundary, was nearly 270 K (orange), and the area of the cloudy pixels, which was nearly 250 K (green Interestingly, in Figure 4b, it can be seen that small clouds were formed from 04:00 to 04:50 UTC, but one dominant cloud which was below 230 K (blue) dramatically increased its size from 04:50 to 05:50 UTC. This phenomenon indicates that the properties of a rapidly developing thunderstorm can be captured by the high-resolution imagery. Therefore the high-resolution imagery accurately can monitor the change of BT11 and be advantages in the detection of early clouds.

**4 Conclusion and limitation**

In this paper, we compared an infrared channel at 10.45 µm of the high-resolution imagery and the low-resolution imagery. It was difficult to track the rapid BT11 changes in the clouds during the process of their development. The lead time of 60 cloud samples determined by the low-resolution imagery was not sufficiently accurately measured to monitor the whole development process of tropical thunderstorms. In contrast, the lead time of the high-resolution imagery was from 90 minutes and to 180 minutes before the cloud reached its mature state. Therefore, the higher spatial and temporal resolution of satellite observations can be valuable as it would alarm for tropical thunderstorms over Southeast Asia approximately two hours earlier than the low-resolution one based on the validation using 60 thunderstorms events.

Some limitations of our study have to be acknowledged. First, this method may not work if a thunderstorm happens in the mesoscale convective system. In the mesoscale convective system, an intrusion of other developing clouds could frequently occur. Second, future studies are needed to determine whether thunderstorms are rainy after the lead time. To more accurately examine the lead time, validation with surface precipitation data based on ground observation is further required. Also, the near-real radar data can be useful to validate the precipitation if the surface precipitation data have not been well managed. Finally, the lead time can differ depending on the region, since the lead time can be affected by various environmental factors such as wind direction and speed, atmospheric profiles, and the characteristics of the geolocation.

The impact of increased spectral observation was not discussed in this study. The additional channels in Himawari-8 are another advantage that improves the accuracy of the forecasts of deep convective clouds, as compared to former satellites. For example, three water vapor channels with different weighting functions can provide more detailed information about the vertical growth of cloud objects. The $CO_2$ absorption channel also gives a chance to monitor the temporal changes in cloud thickness and height. Another infrared channel (8.6 µm) facilitates the examination of the changes in the glaciation of clouds. Therefore, the use of new additional channels in Himawari-8 is critically important to the successful reduction of the false alarm rate in thunderstorm prediction. Implemented operationally under real-life conditions, the high-resolution (2 km and 10 minutes) imager is of great significance to the provision of practical assistance to effective disaster management. The alarm for evacuation can be turned on two hours in advance of the event by using the high-resolution imagery. Although many developing countries in Southeast Asia use Himawari-8 satellite data, few countries carefully consider the importance of satellite resolution. A longer lead time is beneficial to the reduction of the risk of natural disasters caused by thunderstorms and the timely evacuation in such adverse events. Moreover, the low-resolution imagery is more effective in data storage than the low-resolution imagery.

**5 Summary**

Thunderstorm prediction using satellites is of vital importance in Southeast Asian developing countries to reduce the risks from heavy rain, lightning, and flooding. The currently used satellites can observe the area more precisely and their

application to pre-disaster management are highly appreciated and demanded. This study examined the advance in the predictability of thunderstorms using geostationary satellite imageries. Our present results show that by using the latest geostationary satellite data (2 km and 10 minutes resolution), thunderstorms can be predicted 90–180 minutes ahead of their mature state. These data can capture the rapidly growing cloud tops before the cloud moisture fall as precipitation. However,

5 thunderstorms cannot be detected, or can only be detected 60 minutes ahead of time with the low-precision satellite data (4 km and 30 minutes resolution). Therefore, the latest geostationary satellite data obtained at a resolution such as 2 km and 10 minutes can be critically important to the detect early detection of thunderstorms for enabling the prompt preparation and mitigation of hazards.

**Acknowledgements**

10 **This study is supported by the** The Korea Meteorological Administration Research and Development Program under Grant KMI (KMI2018 04110).

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

**Tables**

**Table 1. The number of selected convective clouds during observation time.**

| Month | Observation time | Sample number |
|---|---|---|
| July 2017 | 03:00-06:50 UTC (daytime) | 20 |
| | 21:00-24:50 UTC (nighttime) | 10 |
| August 2017 | 03:00-06:50 UTC (daytime) | 20 |
| | 21:00-24:50 UTC (nighttime) | 10 |

**Table 2. Spatial resolution and time interval of the Advanced Himawari Imager (AHI)/Himawari-8 and MTSAT 1R/2 Imager (JMA/MSC: Himawari-8/9 Imagery (AHI), 2018).**

| | Himawari-8 | | MTSAT 1R/2 | |
|---|---|---|---|---|
| Spatial resolution | Band 3 | 0.5 km | Band 3 | 1 km |
| | Band 1,2,4 | 1 km | | |
| | Band 5-16 | 2 km | Band 7,8,13,15 | 4 km |
| Time interval | Full Disk | 10 min | Full Disk | 30 min |
| | Japan Area and Target Area | 2.5 min | | |
| | Landmark Area | 0.5 min | | |

**Table 3. The lead time according to imagery for cloud No. 1-20 occurred in the daytime of July 2017, No. 21-30 in the nighttime of July 2017, No. 31-50 in the daytime of August 2017, and No. 51-60 in the nighttime of August 2017.**

| No. | Cloud-scale (km) | Lead time (minutes) | | Lead time difference (minutes) (A minus B) |
|---|---|---|---|---|
| | | 2 km and 10 min imagery (A) | 4 km and 30 min imagery (B) | |
| 1 | 120 | 180 | 60 | 120 |
| 2 | 104 | 160 | 30 | 130 |
| 3 | 120 | 140 | 30 | 110 |
| 4 | 120 | 180 | 30 | 150 |
| 5 | 120 | 180 | 60 | 120 |
| 6 | 40 | 130 | 30 | 100 |
| 7 | 40 | 140 | 0 | 140 |
| 8 | 44 | 120 | 0 | 120 |
| 9 | 64 | 120 | 30 | 90 |
| 10 | 40 | 180 | 60 | 120 |
| 11 | 40 | 130 | 0 | 130 |
| 12 | 48 | 90 | 0 | 90 |
| 13 | 96 | 180 | 60 | 120 |
| 14 | 104 | 120 | 0 | 120 |
| 15 | 120 | 140 | 30 | 110 |
| 16 | 80 | 180 | 60 | 120 |
| 17 | 56 | 100 | 0 | 100 |
| 18 | 80 | 180 | 30 | 150 |
| 19 | 96 | 180 | 0 | 180 |
| 20 | 56 | 100 | 0 | 100 |
| 21 | 40 | 160 | 0 | 160 |

| | | | | |
|---|---|---|---|---|
| 22 | 72 | 150 | 30 | 120 |
| 23 | 120 | 140 | 30 | 110 |
| 24 | 96 | 120 | 30 | 90 |
| 25 | 84 | 130 | 0 | 130 |
| 26 | 100 | 180 | 60 | 120 |
| 27 | 60 | 100 | 0 | 100 |
| 28 | 100 | 130 | 0 | 130 |
| 29 | 104 | 120 | 0 | 120 |
| 30 | 96 | 180 | 60 | 120 |
| 31 | 100 | 120 | 30 | 90 |
| 32 | 32 | 100 | 0 | 100 |
| 33 | 48 | 120 | 30 | 90 |
| 34 | 100 | 180 | 30 | 150 |
| 35 | 112 | 180 | 60 | 120 |
| 36 | 64 | 120 | 30 | 90 |
| 37 | 100 | 180 | 60 | 120 |
| 38 | 96 | 140 | 0 | 140 |
| 39 | 68 | 120 | 0 | 120 |
| 40 | 80 | 130 | 30 | 100 |
| 41 | 44 | 90 | 0 | 90 |
| 42 | 60 | 100 | 0 | 100 |
| 43 | 100 | 120 | 0 | 120 |
| 44 | 96 | 120 | 30 | 90 |
| 45 | 68 | 100 | 0 | 100 |
| 46 | 88 | 140 | 30 | 110 |
| 47 | 108 | 180 | 30 | 150 |
| 48 | 124 | 180 | 60 | 120 |
| 49 | 104 | 140 | 0 | 140 |
| 50 | 100 | 120 | 0 | 120 |
| 51 | 120 | 130 | 30 | 100 |
| 52 | 66 | 110 | 0 | 110 |
| 53 | 80 | 130 | 0 | 130 |
| 54 | 120 | 180 | 60 | 120 |
| 55 | 120 | 100 | 0 | 100 |
| 56 | 40 | 180 | 30 | 150 |
| 57 | 56 | 180 | 0 | 180 |
| 58 | 56 | 100 | 0 | 100 |
| 59 | 88 | 160 | 0 | 160 |
| 60 | 92 | 180 | 30 | 150 |

**Figures**

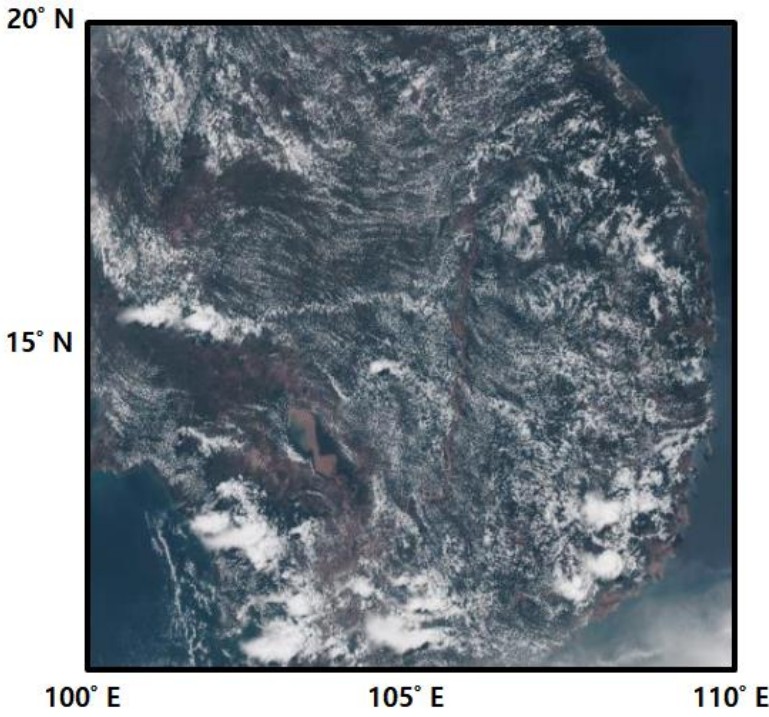

**Figure 1. Himawari-8 AHI RGB image taken for this study area on 19 August 2015, 05:50 UTC. Several convective clouds (white color) are visible in the southern part of the area.**

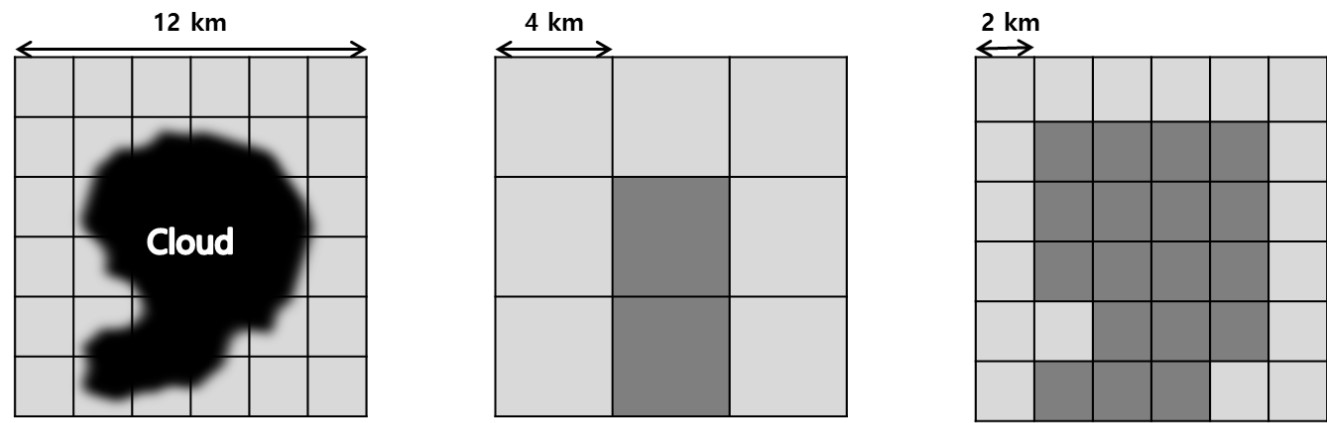

**Figure 2. Illustrations of 12 × 12-km pixels with different resolutions. The dark grey indicates the cloudy pixel, and the light grey indicates the clear-sky pixel. Only 2 cloudy pixels can be detected with the 4-km resolution imagery; in contrast, 18 cloudy pixels can be detected with the 2-km resolution imagery. The number of pixels at the cloud boundary varies depending on the resolution (Walker et al., 2012).**

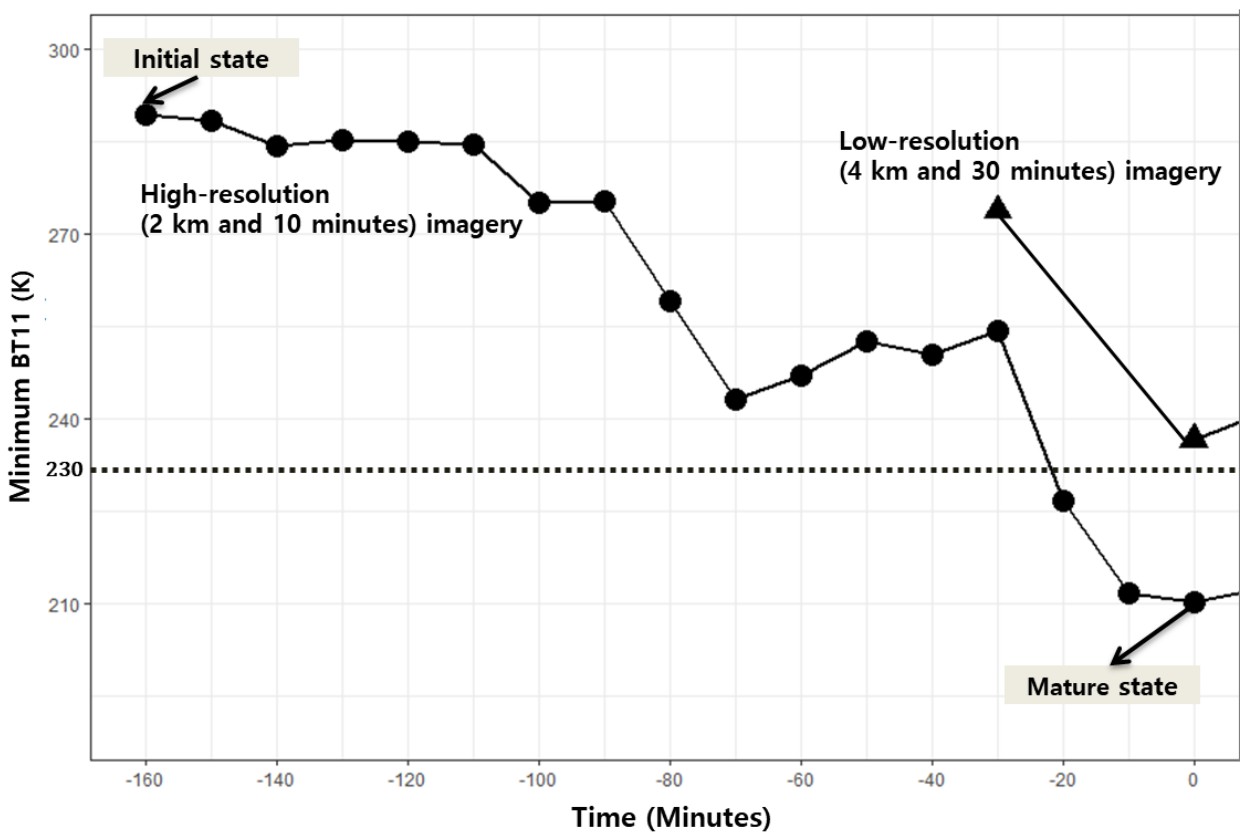

**Figure 3. An example of temporal changes in minimum BT11 among thunderstorm pixels (10 August 2017, 03:10–05:50 UTC) for the high-resolution (2 km and 10 minutes) imagery (circle) and the low-resolution (4 km and 30 minutes) imagery (triangle). In this study, the lead time was defined as the time between the initial state and the mature state (time 0). The negative sign of time indicates the time ahead of the mature state of a tropical thunderstorm.**

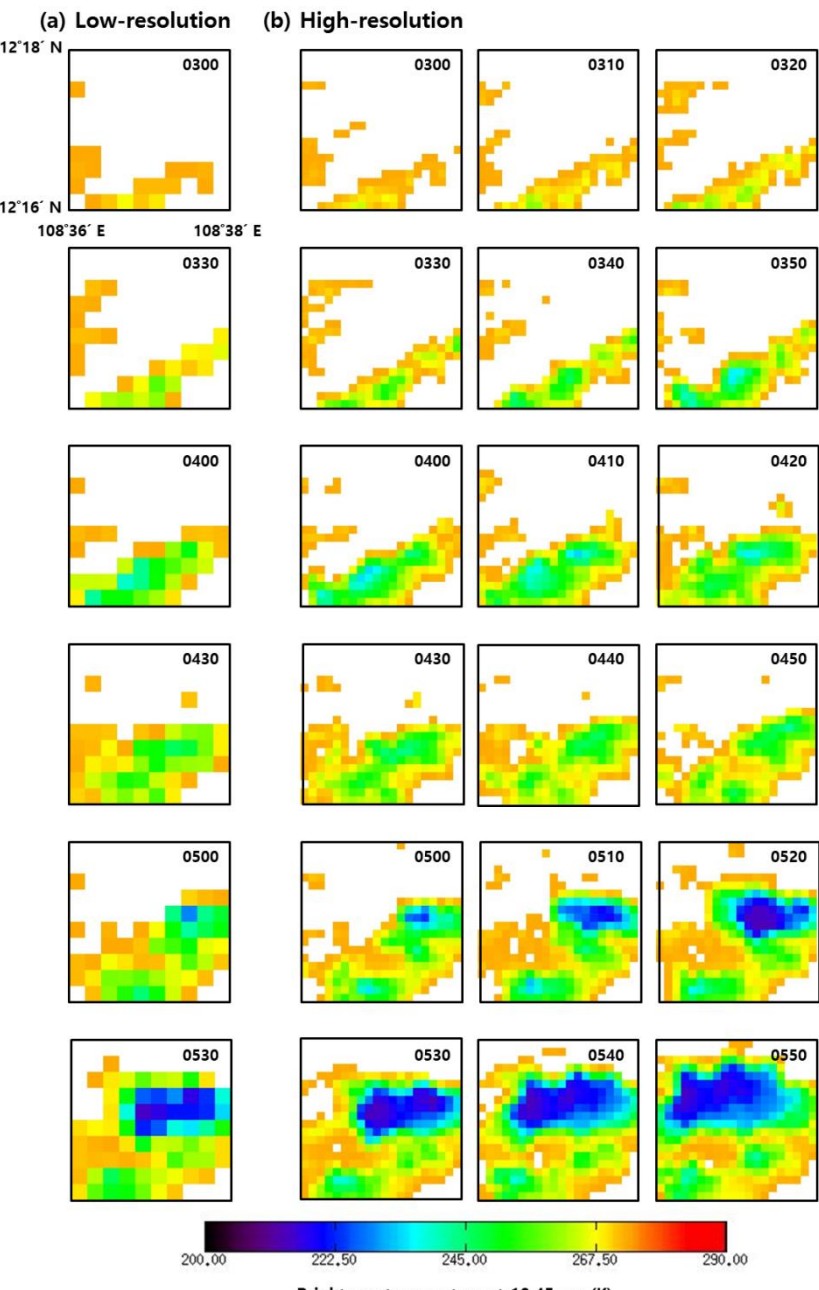

**Figure 4. An example of the thunderstorm development process through BT11 images (10 August 2017, 03:00–05:50 UTC); (a) the low-resolution (4 km and 30 minutes) imagery and (b) the high-resolution (2 km and 10 minutes) imagery.**