# Peer review of "Effect of high-resolution geostationary satellite imageries on predictability of tropical thunderstorms over Southeast Asia"

_Natural Hazards and Earth System Sciences, 2018_

## Short Comment (SC1) · 29 Dec 2018

This paper provides interesting information on predicting thunderstorms by brightness temperature based on geostationary satellite. The importance of time and space resolution is also discussed in detail.

I have some short comments and questions related to this paper:

L15, page 3. How did you convert and smooth data by the time dimension? Is it the average of four 2 km pixels and 30 min?

L26, page 3. Why did you convert brightness temperature to integer?

L4, page 4. The definition of lead time is from initial point to mature point. The initial point is the moment thunder pixels (BT11 decreases more than 5 K in 10 min) are detected. Since the region is 5°-20°N and 100°-110°E, if the thunderstorm like MCS happens near this area and moves in, this method maybe not work. It's better to filter out these examples and expand the data to three months (JJA).

For Figure 2(a), it's better to add longitude and latitude lines. The colorbar looks superfluous at both ends. Maybe you use the colorbar for all examples. But, for the specific one, it's better to set it close to the maximum and minimum of data.

I can't figure out why the lead time of 4 km and 30 min imager is 0 for No. 2. Do you plot the figure of that? Because from 05:00 UTC to 05:40 UTC, many pixels are thunderstorm pixels which should be captured by 4 km and 30 min imager.

Some reference format looks wrong:

L22, page 3: (Schmit et al., 205). L15, page 4: (Houze, 204).

---

## Referee Comment (RC1) · Anonymous Referee #1 · 10 Jan 2019

This paper explores the potential of using brightness temperature from the Himawari-8 satellite to predict tropical thunderstorms compared to lower-resolution images. While a significant improvement in forecasting lead-time is found using the Himawari-8 satellite, the review recommends considering more events (during both daylight and darkness) to evaluate the efficacy of this method. A full thorough English editorial is also recommended prior to publication. Further comments are outlined below.

Major overarching issues:

- The number of events (clouds) considered in this manuscript should be increased. To validate the efficacy of this approach, images from other months and from both

day and night should be considered. The authors' comment about considering daytime only because "the floating population is most active during the daytime", is misinformed. During darkness, many people will be asleep and the lack of direct sunlight may impair the ability of the population to respond to the given hazard effectively. An improvement of lead time during darkness would be a significant contribution to the body of research. As such, it is recommended that both day and night are considered in this analysis.

Specific comments:

- P1 L10: suggest using the term 'significant damage' instead of 'heavy damage'.

- P1 L27: change 'lost' to 'loss'.

- P2 L3: what about storm surge and coastal inundation?

- P2 L11: what do you mean by "grounded"? Do you mean ground-truthed?

- P3 L5: suggest changing "growing" to "developing".

- P3 L6: why is only August considered? This needs to be clarified. The review recommends considering other months in this analysis.

- P3 L22: check the date for Schmit et al., reference.

- P3 L29: change "statuses" to "status".

- P4 L15: check the date for Housze reference.

- P4 L18-L19: check grammar.

- P5 L5: Is "100 min" a typo? Do you mean 10 min?

- P6 L2: This study is validated using a small number of clouds, over a small range, during daylight hours. This needs to be clarified in this sentence.

- Fig 2a: Including coordinates and a continent basemap would be useful.

---

## Author Comment (AC1) · 4 Mar 2019

Dear. open reviewer.

Thanks for your comments and sorry for my late reply. Your sincere comments are helpful to revise this paper. The answers to the questions are as follows.

1) L15, page 3. How did you convert and smooth data by the time dimension? Is it the average of four 2 km pixels and 30 min?

Yes, Himawari-8/9 and MTSAT-1R/2 have different resolution-imager. Therefore, we have to do convert data. It is calculated as the average of four 2 km pixels every 30

min. We called this a virtual MTSAT. Four pixels of 2 km were converted into one pixel of 4 km. Also, the time interval was increased from 10 min to 30 min, which is the same as the MTSAT's spatial resolution and time cycle.

2)L26, page 3. Why did you convert brightness temperature to integer?

Brightness temperature at 10.45 $\mu$m can be expressed as BT11 for readability. The sentence was modified as follows. "Among 16 bands, brightness temperature at 10.45 $\mu$m (hereafter, BT11 for readability) is used for monitoring the vertical growth of clouds."

3)L4, page 4. The definition of lead time is from an initial point to a mature point. The initial point is the moment thunder pixels (BT11 decreases more than 5 K in 10 min) are detected. Since the region is 5âǗę-20âǗęN and 100âǗę-110âǗęE, if the thunderstorm like MCS happens near this area and moves in, this method may not work. It's better to filter out these examples and expand the data to three months (JJA).

Thank you very much for your nice comment. I think we need to think about this in depth. Although many cloud detection technologies have been developed through satellites, it was agreed that improving the satellite's spatial resolution and observation time could have a positive effect, but it was difficult to find a direct quantitative response. The curiosity of this motivated this research. To sum up, this shows that higher spatial and temporal resolutions of satellite observations are more effective for warning people of severe weather from convective clouds over Southeast Asia with the validation using eight clouds, over a small range, during daylight hours. Of course, adding other examples to this paper is good for publicizing the conclusion of the paper. Nevertheless, the conclusion of the paper which suggests the lead time differences due to resolution would be sufficient with cases of night and day since the rapidly developing cloud systems in the tropical region are spatially small and their lifetimes are generally short (< 3 h). I believe that increasing the case will not significantly change the quantitative improvement of early cloud detection, which is the essence of the conclusion.

Since the region is 5°-20°N and 100°-110°E, if the thunderstorm like mesoscale convective system happens near this area and moves in, this method may be useful. But, one thing we should consider is this method is not based on the movement of cloud objects one of the popular methods in mesoscale convective system. If mesoscale convective system could be firstly filtered out from the initial point where BT11 should be over than 270 K, there might be enough problems, such as other developing cloud objects entering the original interested domain. In order to more accurately examine the length of time between the initially detected cumulus and the mature deep convective with heavy rain, validation with precipitation data based on ground observation is further required. Thus, it is premature to assure the lead time for forecasting deep convective clouds over Southeast Asia by this study. In addition, the lead time can differ depending on the region, since the lead time can be affected by various environmental factors such as wind direction and speed, atmospheric profiles, and the characteristics of the geolocation. However, the point that we addressed here is that improved spatial and temporal resolutions of satellites clearly give benefits for enhancing the accuracy of the now-casting forecast.

4)For Figure 2(a), it's better to add longitude and latitude lines. The colorbar looks superfluous at both ends. Maybe you use the colorbar for all examples. But, for the specific one, it's better to set it close to the maximum and minimum of data.

Yes, truly. I'll update it like that. Thank you for telling me a good point!

5)I can't figure out why the lead time of 4 km and 30 min imager is 0 for No. 2. Do you plot the figure of that? Because from 05:00 UTC to 05:40 UTC, many pixels are thunderstorm pixels which should be captured by 4 km and 30 min imager.

*Please look at Fig. 4. (New figure is added for your question.)

As a result, in the cases of the 2 km and 10 min imager, the lead time is from 100 min to 180 min. In contrast, the 4 km and 30 min imager only began to detect cloud pixels up to 30 min before. Of the eight clouds, two made the 30 min prediction, and six clouds

had zero lead time. (Here, a zero lead time means that the 4 km and 30 min imager failed to detect the cloud pixel when the cloud observed at 2 km and 10 min imager reached the mature point.) Figure 4 demonstrates the lead time varies in case of the spatial resolution which is 2 km or 4km and the time interval which is 10 min or 30 min. Each a, b, c shows a change in the minimum value of BT11, that in the average value of BT11 and that in the number of cloud pixels for about 3 hours. The results followed by Figure 4 show the difference of 100 min in lead time when the time interval (10 min) is the same and the spatial resolution (2 km or 4 km) is different, while the difference in lead time depending on time interval (10 min or 30 min) is equal to 30 min when the spatial resolution is equal (4 km). One interesting thing is that the difference in resolution has a greater impact on lead time than on time interval. Especially, as shown in Figure 4c, the spatial resolution has a large impact on the number of cloud pixels. This suggests that the effect of spatial resolution on the observation for early-stage clouds and the near-future cloud development process is very important. At the same time, the impact of improving time intervals cannot be ignored. The reason is the short time cycle, the more precisely the change rate of minimum BT11 is per pixel. For example, only one cloud pixel can be reflected on the area of 16 km in the case of the 4 km resolution imagery; in contrast, four cloud pixels can be reflected on the same area in the case of the 2 km and 10 min imager. In particular, we could not ignore either the influence of the resolution on the boundary of the cloud or the initial stage of cloud growth, when the temperature change is rapid. As the time interval gets shorter, it is possible to observe the movement of the cloud in real time, and this helps the accurate prediction as well as the accumulation of data. In particular, the 4 km and 30 min imager cannot detect initial clouds whose scale is between 2 km and 4 km, so it is difficult to track the whole development process of thunderstorms. Therefore, the prediction of initial clouds through Himawari-8 shows the possibility of detection about two hours earlier than the MTSAT-1R.

6)Some reference format looks wrong: L22, page 3: (Schmit et al., 205). L15, page 4: (Houze, 204).

Yes, these are my mistakes. Schmit et al., 2005 & Houze, 2004 are right.

[Figure]

[Figure]

**Figure 4. Results of detected cloud pixels based on spatial resolution and observation time cycle; (a) Minimum BT11, (b) Mean BT11 and (c) Number of pixels**

**Fig. 1.**

---

## Referee Comment (RC2) · Dieter Poelman (Referee) · 23 May 2019

Manuscript nhess-2018-357 by Lee et al. investigates the effect of increasing resolution of geostationary satellite observations to predict tropical thunderstorms in Southeast Asia. The manuscript fits well within the scope of the NHESS-journal. Although concise, the text is clearly written. However, following points should be addressed before publication:

General comments:
• The goal of this study is very clear. However, I believe that using solely 8 clouds to draw conclusions on the difference in lead time between the old and new observational satellite data is too little. The authors need to include much more data, i.e., increase the amount of observed clouds, simply by extending the amount of observation days (now only 10 & 11 Aug 2017). Why not using for instance all the thunderclouds observed in August 2017 & 2018?
• Fig 1 & 2 only include the data based on the new satellite observations. I believe its worthwhile to include in the figures as well the behaviour of the virtual (lower resolution) data. In this way it is easy for the reader to see the difference between the two.

Specific comments (chronological in appearance):

*1. Introduction:*

• p1, L27, typo: "… lead to extensive economic **losses**"
• p2, L10/11: please rephrase "should be grounded" in this sentence. It is not clear what you mean.
• p2, L16: "(briefly, min)": what do you mean?
• p2, L20/21: "Moreover, … measurements": there are of course advantages using satellite in stead of ground-based observations, but also disadvantages. A network of radars, such as NEXRAD in the US or OPERA in Europe do provide high-resolution precipitation observations over a large area continuously with a higher spatial resolution compared to satellite observations.

*2. Data and Method*

• p2, L30-p3,L2: so is this the only reason why the authors chose this particular region to investigate?
• p3, L6: please rephrase " ...dramatically uprising in the clean sky"
• p3, L11,12: Regions 1, 2 and 3 are mentioned in the text but its not totally clear where those are located. A new figure indicating the three regions would clarify this.
• p3, L14: Reference to JMA/MSC is now "2017", however, in the reference list it is "2018"
• p3, L15: "… whose resolution is **similar to** the MTSAT ..."

*3. Determining thunderstorm pixels and defining the lead time*

• p3, L31: please add a reference(s) for " … BT11 of clouds, which insofar, has been shown to be highly associated with the predictability of thunderstorms [references]"
• p4, L8: please rephrase "… time passed from when the ..." into " time in between the cloud … and ..."
• p4, L15: reference (Houze, 204) → 2004

*4. Improved predictability by comparing lead time differences*

- p4, L25: please rephrase: "the sooner early clouds"
- p4, L26 add references: "Some previous studies have shown … [references]"
- p4, L29: 8 clouds are not a lot to build your conclusions upon
- p5, L2: what is meant with "the floating population"?
- P5, L7/8: remove the brackets "(...)"
- p5, L19: What is meant with "it is difficult to reflect the whole cloud"
- p5, L18-L31: it would be worthwhile to include in Figure 2 the behaviour of the "virtual" lower resolution MTSAT data. In that way, the reader can check visually directly the difference for this particular case.

*5. Conclusions and limitation*

- p6, 1 sentence: this is expected. Even without this study one expects that newer instruments provide higher quality data, which in turn have a positive effect for any meteorological purpose. Exactly what you wrote on p6, L13-15.
- p6, L9-10: please rephrase " … and the mature deep convective with heavy rain"
- p6, L23: "if applied to real technology". Do you mean " if implemented operationally"?
- p6, L26-27: " For example, Cambodia … satellite data are 4 km.": I am wondering if this is really the best place/appropriate (in a scientific article) referring to a specific country. I believe it is better to write in general terms that there are countries in southeast Asia who receive 4km data. For example, last line of the summary is written in more general words, which is in my opinion better.

*Tables:*

I think Table 1 & 3 could be transformed into 1 single table. The observation times in Table 1 can be put into Table 3. However, since more data will be included in the paper, the authors should think how to restructure those tables. I can imagine that when you have not 8, but for example 80 observed clouds it would be better not to use a table but to make a figure of the distribution of the lead times & cloud scales … Is there maybe a relation between the cloud scale and the lead time? More things can be done when including more data!

*Figures:*

- Figure 1: Please rephrase last sentence in the caption of this figure. At the moment, it is not clear. + Include the low-resolution data in this figure as well for direct comparison.
- Figure 2: I would like to see for Fig. 2 a & b that the authors include the virtual (lower resolution) data in order for the reader to see directly the difference between the new and old observational data.

---

## Author Comment (AC2) · 16 Jun 2019

We thank to the reviewer of SC1-supplement for his/her thoughtful comments and clear suggestions. He/she not only indicated the crucial points in our research but also suggested the way how to improve them. Thanks to the comments, the manuscript has been revised as follows. Only eight clouds are a small number to show enough conclusions. We selected clouds that occurred during the day and night in July and August 2017. We added a total of 60 cloud cases, 30 per month.

ïĄň L15, page 3. How did you convert and smooth data by the time dimension? Is it the average of four 2 km pixels and 30 min?

[Figure]

Right. In order to carry out this study, we make virtual data whose resolution is similar to the MTSAT. Specifically, four pixels of 2 km were converted into one pixel of 4 km, and the time interval was increased from 10 minutes to 30 minutes, which is the same as the former spatial resolution and time cycle. In other words, it is calculated as the average of four 2 km pixels every 30 min. To better understating, we added an illustration about the difference in the number of detected cloud pixels by resolution.

ïĄň L26, page 3. Why did you convert brightness temperature to integer?

We simply call the brightness temperature at 10.45 into BT11 for readability. To better understanding, please look at Figure 3.

Please also note the supplement to this comment:
https://www.nat-hazards-earth-syst-sci-discuss.net/nhess-2018-357/nhess-2018-357-AC2-supplement.pdf

---

## Author Comment (AC3) · 16 Jun 2019

Answers to the comments of NHESS-2018-357-RC1-supplement

We thank to the reviewer of RC1-supplement for his/her productive comments and thoughtful guide. He/she not only indicated the crucial points in our research but also suggested the way how to improve them. Thanks to the comments, the manuscript has been revised as follows.

**Major overarching issues:**

- The number of events (clouds) considered in this manuscript should be increased. To validate the efficacy of this approach, images from other months and from both day and night should be considered. The authors' comment about considering daytime only because "the floating population is most active during the daytime" is misinformed. During darkness, many people will be asleep and the lack of direct sunlight may impair the ability of the population to respond to the given hazard effectively. An improvement of lead time during darkness would be a significant contribution to the body of research. As such, it is recommended that both day and night are considered in this analysis.

This is a very helpful comment. Only eight clouds are a small number to show enough conclusions. We selected clouds that occurred during the day and night in July and August 2017. We added a total of 60 cloud cases, 30 per month. Specifically, Table 1 shows information about cloud data. It's a good idea to add 2018 data. However, we didn't have enough time to get the 2018 data. Instead, we used data for two months in 2017 which are July and August, when tropical clouds were frequently observed each year.

**Table 1. The observation time and number of observed clouds in this study.**

| Date | Observation time | Number of observed clouds |
|------|------------------|---------------------------|
| July, 2017 | 03:00 - 06:50 UTC (Day) | 20 |
| | 21:00 - 24:50 UTC (Night) | 10 |
| August, 2017 | 03:00 - 06:50 UTC (Day) | 20 |
| | 21:00 - 24:50 UTC (Night) | 10 |

**Specific comments:**

**The new manuscript has changed the number of lines and pages. The answer below is accompanied by the number of pages and lines.**

- P1 L10: suggest using the term 'significant damage' instead of 'heavy damage'.

Corrected as below

p. 1 line 10 : Tropical thunderstorms cause significant damage to property and lives

- P1 L27: change 'lost' to 'loss'.

Corrected as below

p. 2 line 6-7 : These severe events lead to extensive economic losses, environmental degradation, and subsequently, damage to human life.

- P2 L3: what about storm surge and coastal inundation?

Corrected as below and written the reference

p. 2 line 2-4 : Impacts from recent climate-related extremes, such as heat waves, droughts, floods, cyclones and wildfires, reveal significant vulnerability and exposure of some ecosystems and many human systems to current climate variability (Pachaurim and Meyer, 2014).

- P2 L11: what do you mean by "grounded"? Do you mean ground-truthed?

Removed

- P3 L5: suggest changing "growing" to "developing".

Corrected as below

p. 3 line 19 : convective clouds which are developing within 2 hours

- P3 L6: why is only August considered? This needs to be clarified. The review recommends considering other months in this analysis.

- P6 L2: This study is validated using a small number of clouds, over a small range, during daylight hours. This needs to be clarified in this sentence.

**It is a common answer to the above two questions.**

We added a total of 60 cloud cases. Table 3 shows the result of lead time according to imager; No. 1-20 occurred during the day of July, No. 21-30 occurred during the night of July, No. 31-50 occurred during the day of August and No. 51-60 occurred during the night of August.

Table 3. The lead time according to imagery for cloud No. 1-20 occurred in the daytime of July 2017, No. 21-30 in the nighttime of July 2017, No. 31-50 in the daytime of August 2017, and No. 51-60 in the nighttime of August 2017.

| No. | Cloud scale (km) | Lead time (min) | | Lead time difference (min) (A – B) |
| --- | --- | --- | --- | --- |
| | | 2 km and 10 min imager (A) | 4 km and 30 min imager (B) | |
| 1 | 120 | 180 | 60 | 120 |
| 2 | 104 | 160 | 30 | 130 |
| 3 | 120 | 140 | 30 | 110 |
| 4 | 120 | 180 | 30 | 150 |
| 5 | 120 | 180 | 60 | 120 |
| 6 | 40 | 130 | 30 | 100 |
| 7 | 40 | 140 | 0 | 140 |
| 8 | 44 | 120 | 0 | 120 |
| 9 | 64 | 120 | 30 | 90 |
| 10 | 40 | 180 | 60 | 120 |
| 11 | 40 | 130 | 0 | 130 |
| 12 | 48 | 90 | 0 | 90 |
| 13 | 96 | 180 | 60 | 120 |
| 14 | 104 | 120 | 0 | 120 |
| 15 | 120 | 140 | 30 | 110 |
| 16 | 80 | 180 | 60 | 120 |
| 17 | 56 | 100 | 0 | 100 |
| 18 | 80 | 180 | 30 | 150 |
| 19 | 96 | 180 | 0 | 180 |
| 20 | 56 | 100 | 0 | 100 |
| 21 | 40 | 160 | 0 | 160 |
| 22 | 72 | 150 | 30 | 120 |
| 23 | 120 | 140 | 30 | 110 |
| 24 | 96 | 120 | 30 | 90 |
| 25 | 84 | 130 | 0 | 130 |
| 26 | 100 | 180 | 60 | 120 |
| 27 | 60 | 100 | 0 | 100 |
| 28 | 100 | 130 | 0 | 130 |
| 29 | 104 | 120 | 0 | 120 |
| 30 | 96 | 180 | 60 | 120 |
| 31 | 100 | 120 | 30 | 90 |
| 32 | 32 | 100 | 0 | 100 |
| 33 | 48 | 120 | 30 | 90 |
| 34 | 100 | 180 | 30 | 150 |
| 35 | 112 | 180 | 60 | 120 |
| 36 | 64 | 120 | 30 | 90 |
| 37 | 100 | 180 | 60 | 120 |
| 38 | 96 | 140 | 0 | 140 |
| 39 | 68 | 120 | 0 | 120 |
| 40 | 80 | 130 | 30 | 100 |
| 41 | 44 | 90 | 0 | 90 |
| 42 | 60 | 100 | 0 | 100 |
| 43 | 100 | 120 | 0 | 120 |
| 44 | 96 | 120 | 30 | 90 |
| 45 | 68 | 100 | 0 | 100 |
| 46 | 88 | 140 | 30 | 110 |
| 47 | 108 | 180 | 30 | 150 |

| 48 | 124 | 180 | 60 | 120 |
|----|-----|-----|-----|-----|
| 49 | 104 | 140 | 0 | 140 |
| 50 | 100 | 120 | 0 | 120 |
| 51 | 120 | 130 | 30 | 100 |
| 52 | 66 | 110 | 0 | 110 |
| 53 | 80 | 130 | 0 | 130 |
| 54 | 120 | 180 | 60 | 120 |
| 55 | 120 | 100 | 0 | 100 |
| 56 | 40 | 180 | 30 | 150 |
| 57 | 56 | 180 | 0 | 180 |
| 58 | 56 | 100 | 0 | 100 |
| 59 | 88 | 160 | 0 | 160 |
| 60 | 92 | 180 | 30 | 150 |

- P3 L22: check the date for Schmit et al., reference.

Corrected as below

p. 4, line 6 : (Schmit et al., 2005).

- P3 L29: change "statuses" to "status".

Corrected as below

p. 4, line 14 : the current status of clouds

- P4 L15: check the date for Housze reference.

Corrected as below

p. 4, line 11 : (Houze, 2004)

- P4 L18-L19: check grammar.

"grammar check is completed"

p. 5, line 6-7 : The interesting point of Figure 2 is the pattern of temporal changes in minimum BT11 among the thunderstorm pixels for high-resolution imagery. One can expect that BT11 of a thunderstorm might gradually decrease, but the BT11 of the targeted thunderstorm firstly decrease from the initial state to −70 minutes and increase slightly from −70 minutes to −30 minutes. We inferred that the decline of BT11 relates to the vertical growth of the cloud, while the increase of BT11 after reaching the mature state relates to the horizontal expansions of the cloud. This is commonly observed in the life cycle of tropical thunderstorms. It is notable that BT11 for low-resolution imagery is too simple to monitor the status of clouds in detail.

- P5 L5: Is "100 min" a typo? Do you mean 10 min?

10 minutes

- Fig 2a: Including coordinates and a continent basemap would be useful.

[Figure]

**Figure 1. Himawari-8 AHI RGB image taken for this study area on 19 August 2015, 05:50 UTC. Several convective clouds (white color) are shown in the southern part of the area**

---

## Author Comment (AC4) · 17 Jun 2019

We thank the reviewer of RC2-supplement for his/her productive comments and keen insight. He/she not only indicated the crucial points in our research but also suggested the way how to improve them. Thanks to the comments, the manuscript has been revised as follows.

Please open the attached file and you can check all answers for the comments of NHESS-2018-357-RC2-supplement.

Please also note the supplement to this comment:

[Figure]

https://www.nat-hazards-earth-syst-sci-discuss.net/nhess-2018-357/nhess-2018-357-AC4-supplement.pdf

[Figure]

**Supplement:**

Answers to the comments of NHESS-2018-357-RC2-supplement

We thank to reviewer of RC2-supplement for his/her productive comments and keen insight. He/she not only indicated the crucial points in our research but also suggested the way how to improve them. Thanks to the comments, the manuscript has been revised as follows.

**General comments:**

• The goal of this study is very clear. However, I believe that using solely 8 clouds to draw conclusions on the difference in lead time between the old and new observational satellite data is too little. The authors need to include much more data, i.e., increase the amount of observed clouds, simply by extending the amount of observation days (now only 10 & 11 Aug 2017). Why not using for instance all the thunderclouds observed in August 2017 & 2018?

Only eight clouds are a small number to show enough conclusions. We selected clouds that occurred during the day and night in July and August 2017. We added a total of 60 cloud cases, 30 per month. Specifically, Table 1 shows information about cloud data. It's a good idea to add 2018 data. However, we didn't have enough time to get the 2018 data. Instead, we used data for two months in 2017 which are July and August, when tropical clouds were frequently observed each year.

p. 10

**Table 1. The observation time and number of observed clouds in this study.**

| Date | Observation time | Number of observed clouds |
|---|---|---|
| July, 2017 | 03:00 - 06:50 UTC (Day) | 20 |
| | 21:00 -   24:50 UTC (Night) | 10 |
| August, 2017 | 03:00 - 06:50 UTC (Day) | 20 |
| | 21:00 - 24:50 UTC (Night) | 10 |

• Fig 1 & 2 only include the data based on the new satellite observations. I believe its worthwhile to include in the figures as well the behavior of the virtual (lower resolution) data. In this way it is easy for the reader to see the difference between the two.

I totally agree with this comment. Figures have revised for the reader to compare high-resolution and low-resolution data.

p. 13

[Figure]

Figure 2. An example of temporal changes in minimum BT11 among thunderstorm pixels (10 August 2017, 03:10-05:50 UTC) for the high-resolution (2 km and 10 minutes) imagery (circle) and the low-resolution (4 km and 30 minutes) imagery (triangle). In this study, the lead time is defined as the time between the initial state and the mature state (time 0). The negative sign of time indicates the time ahead of the mature state of a tropical thunderstorm

[Figure]

**Figure 3. An example of the thunderstorm development process through BT11 images (10 August 2017, 03:00-05:50 UTC); (a) the low-resolution (4 km and 30 minutes) imagery and (b) the high-resolution (2 km and 10 minutes) imagery**

[Figure]

Figure 4. Illustrations of 12 × 12 km pixels with different resolutions. The dark grey indicates the cloudy pixel, and the light grey indicates the clear-sky pixel. Only 2 cloudy pixels can be detected with the 4 km resolution imagery; in contrast, 18 cloudy pixels can be detected with the 2 km resolution imagery. The number of pixels at the cloud boundary varies depending on the resolution.

**Specific comments (chronological in appearance):**

**The new manuscript has changed the number of lines and pages.**

**1. Introduction:**

• p1, L27, typo: "… lead to extensive economic losses"

Corrected as below

p. 2, line 6

These severe events lead to extensive economic losses, environmental degradation, and subsequently, damage to human life.

• p2, L10/11: please rephrase "should be grounded" in this sentence. It is not clear what you mean.

Removed

• p2, L16: "(briefly, min)": what do you mean?

Removed.

• p2, L20/21: "Moreover, … measurements": there are of course advantages using satellite in stead of ground-based observations, but also disadvantages. A network of radars, such as NEXRAD in the US or OPERA in Europe do provide high-resolution precipitation observations over a large area continuously with a higher spatial

resolution compared to satellite observations.

This is very helpful comment. We added to the manuscript that some countries are using high-resolution radar system as well as satellites as follows.

p. 2, line 14 – 19

Not only is the model itself insufficient, but the observational data to support the modeling are also insufficient. To make matters worse, unlike the middle latitudes, the tropical atmosphere is conditionally unstable, making the models hard to predict tropical thunderstorms. Hence, alarms for the hazards in the tropics are generally managed by the nowcasting system by real-time observations from radar and meteorological satellites. For example, European Operational Program for Exchange of Weather Radar Information (OPERA) do provide precipitation data over a large area continuously with a higher spatial resolution compared to satellite observations (Huuskonen et al., 2014).

**2. Data and Method**

• p2, L30-p3,L2: so is this the only reason why the authors chose this particular region to investigate?

[Figure]

**Figure 1. Himawari-8 AHI RGB image taken for this study area on 19 August 2015, 05:50 UTC. Several convective clouds (white color) are shown in the southern part of the area**

p. 3, line 4 – 12

The region of interest of this study is from 10°N to 20°N and from 100°E to 120°E, which is closely related to the Mekong River Commission. The Mekong River Commission is the only inter-governmental organization that works directly with the governments of Cambodia, Lao PDR, Thailand, and Viet Nam to jointly manage the shared water resources and the sustainable development of the Mekong River (Jacobs, 2002). Unfortunately, this is known as a vulnerable disaster region because of a high risk of extreme weather. As changes in weather patterns are being felt across the Mekong River Commission, the impacts of climate change have become a strong issue. The warmer atmosphere can contain more moisture, which increases the potential to invigorate thunderstorms if all else being equal. It is generally assumed that the temperature increase associated with global climate change will lead to increased thunderstorm intensity and associated heavy precipitation events (Schefczyk et al. , 2015).

• p3, L6: please rephrase " ...dramatically uprising in the clean sky"

"Corrected as below"

p. 3 line 23

convective clouds which are developing within 2 hours in the clear sky

• p3, L11,12: Regions 1, 2 and 3 are mentioned in the text but its not totally clear where those are located. A new figure indicating the three regions would clarify this.

The observation range and time interval vary for each area in Himawari-8 AHI observation. In Southeast Asia, it does not belong to specific targeted regions which are from Region 1 and Region 5.like below.

[Figure]

Thus, this study only observed the Southeast Asia, high-resolution data provided per 10-minutes were used. Different time cycles of observation for each region are not covered in detail.

• p3, L14: Reference to JMA/MSC is now "2017", however, in the reference list it is "2018"

Corrected as below

p. 3, line 28

(JMA/MSC: Himawari-8/9 Imager (AHI), 2018).

• p3, L29: "… whose resolution is similar to the MTSAT ..."

Corrected as below

p. 4, line 1

In order to carry out this study, we make virtual data whose resolution is similar to the MTSAT

**3. Determining thunderstorm pixels and defining the lead time**

• p3, L31: please add a reference(s) for " … BT11 of clouds, which insofar, has been shown to be highly associated with the predictability of thunderstorms [references]"

References are written and rearrange the paragraph

p. 4, line 10-15

60 thunderstorms are subjectively selected based on the RGB images over Southeast Asia. Sizes of selected thunderstorms are less than 120 km since those convective-scales typically accompany precipitation (Houze, 2004). We set the target boundaries depending on the thunderstorm size and the specific locations of thunderstorms are different. In that target boundary, BT11 values are monitored to determine thunderstorm pixels and phases (initial/mature states) for the whole life cycle of thunderstorms. It is because temporal changes in BT11 inform vertical drift velocity, the current status of clouds and diagnose the probability of imminent heavy rains/lightning soon (Vila et al., 2008).

• p4, L8: please rephrase "… time passed from when the ..." into " time in between the

cloud … and ..."

Corrected as below

p. 4, line 24

The lead time is defined as the time between the initial state and the mature state.

• p4, L15: reference (Houze, 204) → 2004

Corrected as below

p. 5, line 15

(Houze, 2004)

**4. Improved predictability by comparing lead time differences**

• p4, L25: please rephrase: "the sooner early clouds"

Rephrased as below

p. 7, line 2 - 3

To reduce the risk of natural disasters after occurring thunderstorms, the long lead time is beneficial to disaster risk reduction.

• p4, L26 add references: "Some previous studies have shown … [references]"

Corrected as below

p. 4, line 10 - 11

60 thunderstorms are subjectively selected based on the RGB images over Southeast Asia. Sizes of selected thunderstorms are less than 120 km since those convective-scales typically accompany precipitation (Houze, 2004).

• p4, L29: 8 clouds are not a lot to build your conclusions upon

We added a total of 60 cloud cases. Table 3 shows the result of lead time according to imagery.

**Table 3. The lead time according to imagery for cloud No. 1-20 occurred in the daytime of July 2017, No. 21-30 in the nighttime of July 2017, No. 31-50 in the daytime of August 2017, and No. 51-60 in the nighttime of August 2017.**

| No. | Cloud-scale (km) | Lead time (min) | | Lead time difference (min) (A minus B) |
|---|---|---|---|---|
| | | 2 km and 10 min imagery (A) | 4 km and 30 min imagery (B) | |
| 1 | 120 | 180 | 60 | 120 |
| 2 | 104 | 160 | 30 | 130 |
| 3 | 120 | 140 | 30 | 110 |
| 4 | 120 | 180 | 30 | 150 |
| 5 | 120 | 180 | 60 | 120 |
| 6 | 40 | 130 | 30 | 100 |
| 7 | 40 | 140 | 0 | 140 |
| 8 | 44 | 120 | 0 | 120 |
| 9 | 64 | 120 | 30 | 90 |
| 10 | 40 | 180 | 60 | 120 |
| 11 | 40 | 130 | 0 | 130 |
| 12 | 48 | 90 | 0 | 90 |
| 13 | 96 | 180 | 60 | 120 |
| 14 | 104 | 120 | 0 | 120 |
| 15 | 120 | 140 | 30 | 110 |
| 16 | 80 | 180 | 60 | 120 |
| 17 | 56 | 100 | 0 | 100 |
| 18 | 80 | 180 | 30 | 150 |
| 19 | 96 | 180 | 0 | 180 |

| 20 | 56 | 100 | 0 | 100 |
|---|---|---|---|---|
| 21 | 40 | 160 | 0 | 160 |
| 22 | 72 | 150 | 30 | 120 |
| 23 | 120 | 140 | 30 | 110 |
| 24 | 96 | 120 | 30 | 90 |
| 25 | 84 | 130 | 0 | 130 |
| 26 | 100 | 180 | 60 | 120 |
| 27 | 60 | 100 | 0 | 100 |
| 28 | 100 | 130 | 0 | 130 |
| 29 | 104 | 120 | 0 | 120 |
| 30 | 96 | 180 | 60 | 120 |
| 31 | 100 | 120 | 30 | 90 |
| 32 | 32 | 100 | 0 | 100 |
| 33 | 48 | 120 | 30 | 90 |
| 34 | 100 | 180 | 30 | 150 |
| 35 | 112 | 180 | 60 | 120 |
| 36 | 64 | 120 | 30 | 90 |
| 37 | 100 | 180 | 60 | 120 |
| 38 | 96 | 140 | 0 | 140 |
| 39 | 68 | 120 | 0 | 120 |
| 40 | 80 | 130 | 30 | 100 |
| 41 | 44 | 90 | 0 | 90 |
| 42 | 60 | 100 | 0 | 100 |
| 43 | 100 | 120 | 0 | 120 |
| 44 | 96 | 120 | 30 | 90 |
| 45 | 68 | 100 | 0 | 100 |
| 46 | 88 | 140 | 30 | 110 |
| 47 | 108 | 180 | 30 | 150 |
| 48 | 124 | 180 | 60 | 120 |
| 49 | 104 | 140 | 0 | 140 |
| 50 | 100 | 120 | 0 | 120 |
| 51 | 120 | 130 | 30 | 100 |
| 52 | 66 | 110 | 0 | 110 |
| 53 | 80 | 130 | 0 | 130 |
| 54 | 120 | 180 | 60 | 120 |
| 55 | 120 | 100 | 0 | 100 |
| 56 | 40 | 180 | 30 | 150 |
| 57 | 56 | 180 | 0 | 180 |
| 58 | 56 | 100 | 0 | 100 |
| 59 | 88 | 160 | 0 | 160 |
| 60 | 92 | 180 | 30 | 150 |

• p5, L2: what is meant with "the floating population"?

Removed

• P5, L7/8: remove the brackets "(...)"

Removed

• p5, L19: What is meant with "it is difficult to reflect the whole cloud"

Corrected as below

p. 5, line 27-31

To monitor the rapid development of the thunderstorm within 2 hours, it is better to consider the influence of the resolution on the boundary of the clouds. The higher the resolution, the more precisely the change rate of minimum BT11 is per pixel. The low-resolution imagery cannot detect cloudy pixels whose scale is between 2 km and 4 km. For example, only one cloud pixel can be reflected on the area of 16 km2 in the case of 4 km resolution imagery per 30 minutes; whereas, four cloud pixels can be reflected on the same area in the case of 2 km imagery per 10 minutes.

• p5, L18-L31: it would be worthwhile to include in Figure 2 the behaviour of the "virtual" lower resolution MTSAT data. In that way, the reader can check visually directly the difference for this particular case.

"Yes, that's good point. Figure 2 is modified."

[Figure]

**Figure 2. An example of temporal changes in minimum BT11 among thunderstorm pixels (10 August 2017, 03:10-05:50 UTC) for the high-resolution (2 km and 10 minutes) imagery (circle) and the low-resolution (4 km and 30 minutes) imagery (triangle). In this study, the lead time is defined as the time between the initial state and the mature state (time 0). The negative sign of time indicates the time ahead of the mature state of a tropical thunderstorm**

**5. Conclusions and limitation**

• p6, 1 sentence: this is expected. Even without this study one expects that newer instruments provide higher quality data, which in turn have a positive effect for any meteorological purpose. Exactly what you wrote on p6, L13-15.

We have clearly rearranged the paragraph.

p. 6 line 9 - 15

In this paper, we compared one infrared channel at 10.45 µm of the high-resolution imagery and the low-resolution imagery. It is difficult to track rapid BT11 changes of clouds in the development process. As a result of 60 cloud samples, the lead time with the low-resolution imagery is not enough to monitor the whole development process of tropical thunderstorms because the maximum of lead time is 60 minutes. In contrast, the lead time of high-resolution imagery is from 90 minutes and 180 minutes before the cloud reach to mature state. Therefore, this shows that higher spatial and temporal resolution of satellite observations can be more effective for alarming about 2 hours earlier tropical thunderstorms over Southeast Asia with the validation using 60 thunderstorms events.

• p6, L9-10: please rephrase " … and the mature deep convective with heavy rain"

Corrected as below

p. 6 line 18-20

Second, future studies are needed to determine whether thunderstorms are rainy after lead time. In order to more accurately examine the lead time, validation with surface precipitation data based on ground observation is further required.

• p6, L23: "if applied to real technology". Do you mean " if implemented operationally?"

Corrected as below

p. 6 line 29-30

If implemented operationally to real life, the high-resolution (2 km and 10 minutes) imager is required to provide practical assistance to disaster management.

• p6, L26-27: " For example, Cambodia … satellite data are 4 km.": I am wondering if this is really the best place/appropriate (in a scientific article) referring to a specific country. I believe it is better to write in general terms that there are countries in southeast Asia who receive 4km data. For example, last line of the summary is written in more general words, which is in my opinion better.

Absolutely, you are right. Thanks for your sincere comment!

p. 6 line 30 - 32

With the high-resolution imagery, the alarm for evacuation can disseminate two hours in advance. Although there are many developing countries in Southeast Asia using the Himawari-8 satellite data, few countries carefully think about satellite resolution.

Tables:

I think Table 1 & 3 could be transformed into 1 single table. The observation times in Table 1 can be put into Table 3. However, since more data will be included in the paper, the authors should think how to restructure those tables. I can imagine that when you have not 8, but for example 80 observed clouds it would be better not to use a table but to make a figure of the distribution of the lead times & cloud scales … Is there maybe a relation between the cloud scale and the lead time? More things can be done when including more data!

60 cloud examples are difficult to represent in a single table. The results are divided into Tables 1 and 3.

**Table 1. the sample number obtained during each observation time.**

| Month | Observation time | Sample number |
|---|---|---|
| July 2017 | 03:00-06:50 UTC (Day) | 20 |
| | 21:00-24:50 UTC (Night) | 10 |
| August 2017 | 03:00-06:50 UTC (Day) | 20 |
| | 21:00-24:50 UTC (Night) | 10 |

At the same time, thank you for suggesting new ideas related to this study. To analyze the correlation between cloud size and lead time, a preliminary test was conducted with 60 cloud samples. As a result, it was difficult to see a notable correlation between cloud scale and lead time. We consider that the number of samples will be increased to identify the relationship between cloud size and lead time. But, your idea is worth studying in the future. More detailed assumptions and definition about cloud scale will be needed to precede the research.

(a) Correlation : 0.3755376

(b) Correlation : 0.3755376

[Figure]

**Figure. The correlation between lead time and cloud scale; (a) the high-resolution (2 km and 10 minutes) imagery and (b) the low-resolution (4 km and 30 minutes) imagery**

Figures:

• Figure 1: Please rephrase last sentence in the caption of this figure. At the moment, it is not clear. + Include the low-resolution data in this figure as well for direct comparison.

That's good. Sentences in the caption of Figure 2 are revised.

[Figure]

**Figure 2. An example of temporal changes in minimum BT11 among thunderstorm pixels (10 August 2017, 03:10-05:50 UTC) for the high-resolution (2 km and 10 minutes) imagery (circle) and the low-resolution (4 km and 30 minutes) imagery (triangle). In this study, the lead time is defined as the time between the initial state and the mature state (time 0). The negative sign of time indicates the time ahead of the mature state of a tropical thunderstorm**

• Figure 2: I would like to see for Fig. 2 a & b that the authors include the virtual (lower resolution) data in order for the reader to see directly the difference between the new and old observational data.

"Yes, that's right. Figure 3 is modified for direct comparison of imageries."

[Figure]

**Figure 3. An example of the thunderstorm development process through BT11 images (10 August 2017, 03:00-05:50 UTC); (a) the low-resolution (4 km and 30 minutes) imagery and (b) the high-resolution (2 km and 10 minutes) imagery**

---

## Author Response (AR1)

**Authors response-point by point**

Answers to the comments of NHESS-2018-357-SC1-supplement

We thank to reviewer of SC1-supplement for his/her thoughtful comments and clear suggestions. He/she not only indicated the crucial points in our research but also suggested the way how to improve them. Thanks to the comments, the manuscript has been revised as follows.

- L15, page 3. How did you convert and smooth data by the time dimension? Is it the average of four 2 km pixels and 30 min?

"Rephrased"

P 4 line 7-20

To perform this study, we created virtual data whose resolution was similar to those of MTSAT 1R/2 (Table 2). Specifically, four pixels of 2 km were converted into one pixel of 4 km, and the time interval was increased from 10 minutes to 30 minutes. In other words, it was calculated as the average of four 2-km pixels in the process of observing clouds every 30 minutes. The number of detected cloudy pixels by resolution is illustrated in Figure 2. A tropical thunderstorm was found to be located in the area of 12 km. The dark grey pixels indicate the ones detected, whereas clouds and the light grey pixels indicate clear-sky pixels. Using the 4-km resolution imagery only 2 cloudy pixels were detected in the middle area with the 4- km resolution imagery; in contrast, 18 cloudy pixels can be detected with the 2-km resolution imagery. It is noteworthy that the high-resolution imagery was able to detect cloudy pixels located at a curved boundary. However, the low-resolution imagery tended to simplify the change rate of minimum BT11, and the detection of cloudy pixels at a curved boundary was somewhat hard (Walker et al., 2012). Hereafter, the virtual MTSAT is called the low-resolution (4 km and 30 minutes) imagery and the Himawari-8 is called the high-resolution (2 km and 10 minutes) imagery so as to facilitate the intuitive understanding of the spatiotemporal resolution difference. Our final study aim was to quantitatively establish the high-resolution imagery low-resolution imagery and compare their effectiveness in the advanced predictability of tropical thunderstorms.

[Figure]

**Figure 2. Illustrations of 12 × 12-km pixels with different resolutions. The dark grey indicates the cloudy pixel, and the light grey indicates the clear-sky pixel. Only 2 cloudy pixels can be detected with the 4-km resolution imagery; in contrast, 18 cloudy pixels can be detected with the 2-km resolution imagery. The number of pixels at the cloud boundary varies depending on the resolution (Walker et al., 2012).**

- L26, page 3. Why did you convert brightness temperature to integer?

"We simply call the brightness temperature at 10.45 into BT11 for readability." Among the 16 existing bands, the brightness temperature at 10.45 µm (BT11) was used for monitoring the vertical growth of clouds.

Answers to the comments of NHESS-2018-357-RC1-supplement

We thank to the reviewer of RC1-supplement for his/her productive comments and thoughtful guide. He/she not only indicated the crucial points in our research but also suggested the way how to improve them. Thanks to the comments, the manuscript has been revised as follows.

**Major overarching issues:**

- The number of events (clouds) considered in this manuscript should be increased. To validate the efficacy of this approach, images from other months and from both day and night should be considered. The authors' comment about considering daytime only because "the floating population is most active during the daytime" is misinformed. During darkness, many people will be asleep and the lack of direct sunlight may impair the ability of the population to respond to the given hazard effectively. An improvement of lead time during darkness would be a significant contribution to the body of research. As such, it is recommended that both day and night are considered in this analysis.

"This is a very helpful comment. Only eight clouds are a small number to show enough conclusions. We selected clouds that occurred during the day and night in July and August 2017. We added a total of 60 cloud cases, 30 per month. Specifically, Table 1 shows information about cloud data. It's a good idea to add 2018 data. However, we didn't have enough time to get the 2018 data. Instead, we used data for two months in 2017 which are July and August, when tropical clouds were frequently observed each year."

P 10

**Table 1. The number of selected convective clouds during observation time.**

| Month | Observation time | Sample number |
|---|---|---|
| July 2017 | 03:00-06:50 UTC (daytime) | 20 |
| | 21:00-24:50 UTC (nighttime) | 10 |
| August 2017 | 03:00-06:50 UTC (daytime) | 20 |
| | 21:00-24:50 UTC (nighttime) | 10 |

**Specific comments:**

- P1 L10: suggest using the term 'significant damage' instead of 'heavy damage'.

"Tropical thunderstorms cause significant damage to property and lives, and a strong research interest exists in and efforts have been focused on the advance and improvement of the thunderstorm predictability by s atellite observations."

- P1 L27: change 'lost' to 'loss'.

"These severe events cause extensive economic losses, environmental degradation, and subsequent damage to human life."

- P2 L3: what about storm surge and coastal inundation?

"Impacts from recent climate-related extremes, such as heat waves, droughts, floods, cyclones, and wildfires, reveal   the significant vulnerability and exposure of some ecosystems and many human systems to the current climate variability (Pachauri and Meyer, 2014)."

- P2 L11: what do you mean by "grounded"? Do you mean ground-truthed?

"Removed"

- P3 L5: suggest changing "growing" to "developing".

"Changed"

- P3 L6: why is only August considered? This needs to be clarified. The review recommends considering other months in this analysis.

- P6 L2: This study is validated using a small number of clouds, over a small range, during daylight hours. This needs to be clarified in this sentence.

"We added a total of 60 cloud cases. Table 3 includes the result of updated analysis"

P 10

**Table 3. The lead time according to imagery for cloud No. 1-20 occurred in the daytime of July 2017, No. 21-30 in the nighttime of July 2017, No. 31-50 in the daytime of August 2017, and No. 51-60 in the nighttime of August 2017.**

| No. | Cloud-scale (km) | Lead time (minutes) | | Lead time difference (minutes) (A minus B) |
|---|---|---|---|---|
| | | 2 km and 10 min imagery (A) | 4 km and 30 min imagery (B) | |
| 1 | 120 | 180 | 60 | 120 |
| 2 | 104 | 160 | 30 | 130 |
| 3 | 120 | 140 | 30 | 110 |
| 4 | 120 | 180 | 30 | 150 |
| 5 | 120 | 180 | 60 | 120 |
| 6 | 40 | 130 | 30 | 100 |
| 7 | 40 | 140 | 0 | 140 |
| 8 | 44 | 120 | 0 | 120 |
| 9 | 64 | 120 | 30 | 90 |

| | | | | |
|---|---|---|---|---|
| 10 | 40 | 180 | 60 | 120 |
| 11 | 40 | 130 | 0 | 130 |
| 12 | 48 | 90 | 0 | 90 |
| 13 | 96 | 180 | 60 | 120 |
| 14 | 104 | 120 | 0 | 120 |
| 15 | 120 | 140 | 30 | 110 |
| 16 | 80 | 180 | 60 | 120 |
| 17 | 56 | 100 | 0 | 100 |
| 18 | 80 | 180 | 30 | 150 |
| 19 | 96 | 180 | 0 | 180 |
| 20 | 56 | 100 | 0 | 100 |
| 21 | 40 | 160 | 0 | 160 |
| 22 | 72 | 150 | 30 | 120 |
| 23 | 120 | 140 | 30 | 110 |
| 24 | 96 | 120 | 30 | 90 |
| 25 | 84 | 130 | 0 | 130 |
| 26 | 100 | 180 | 60 | 120 |
| 27 | 60 | 100 | 0 | 100 |
| 28 | 100 | 130 | 0 | 130 |
| 29 | 104 | 120 | 0 | 120 |
| 30 | 96 | 180 | 60 | 120 |
| 31 | 100 | 120 | 30 | 90 |
| 32 | 32 | 100 | 0 | 100 |
| 33 | 48 | 120 | 30 | 90 |
| 34 | 100 | 180 | 30 | 150 |
| 35 | 112 | 180 | 60 | 120 |
| 36 | 64 | 120 | 30 | 90 |
| 37 | 100 | 180 | 60 | 120 |
| 38 | 96 | 140 | 0 | 140 |
| 39 | 68 | 120 | 0 | 120 |
| 40 | 80 | 130 | 30 | 100 |
| 41 | 44 | 90 | 0 | 90 |
| 42 | 60 | 100 | 0 | 100 |
| 43 | 100 | 120 | 0 | 120 |
| 44 | 96 | 120 | 30 | 90 |
| 45 | 68 | 100 | 0 | 100 |
| 46 | 88 | 140 | 30 | 110 |
| 47 | 108 | 180 | 30 | 150 |
| 48 | 124 | 180 | 60 | 120 |
| 49 | 104 | 140 | 0 | 140 |
| 50 | 100 | 120 | 0 | 120 |
| 51 | 120 | 130 | 30 | 100 |
| 52 | 66 | 110 | 0 | 110 |
| 53 | 80 | 130 | 0 | 130 |
| 54 | 120 | 180 | 60 | 120 |
| 55 | 120 | 100 | 0 | 100 |
| 56 | 40 | 180 | 30 | 150 |
| 57 | 56 | 180 | 0 | 180 |
| 58 | 56 | 100 | 0 | 100 |
| 59 | 88 | 160 | 0 | 160 |
| 60 | 92 | 180 | 30 | 150 |

- P3 L22: check the date for Schmit et al., reference.

"Correted" (Schmit et al., 2005).

- P3 L29: change "statuses" to "status"

"Changed"

- P4 L15: check the date for Housze reference.

"Correted" (Houze Jr, 2004)

- P4 L18-L19: check grammar.

"grammar check is completed"

- P5 L5: Is "100 min" a typo? Do you mean 10 min?

"10 minutes"

- Fig 2a: Including coordinates and a continent basemap would be useful.

"Figure 1 indicated the region of interest in this study including coordinates"

[Figure]

**Figure 1. Himawari-8 AHI RGB image taken for this study area on 19 August 2015, 05:50 UTC. Several convective clouds (white color) are visible in the southern part of the area.**

Answers to the comments of NHESS-2018-357-RC2-supplement

We thank to reviewer of RC2-supplement for his/her productive comments and keen insight. He/she not only indicated the crucial points in our research but also suggested the way how to improve them. Thanks to the comments, the manuscript has been revised as follows.

**General comments:**

• The goal of this study is very clear. However, I believe that using solely 8 clouds to draw conclusions on the difference in lead time between the old and new observational satellite data is too little. The authors need to include much more data, i.e., increase the amount of observed clouds, simply by extending the amount of observation days (now only 10 & 11 Aug 2017). Why not using for instance all the thunderclouds observed in August 2017 & 2018?

"Only eight clouds are a small number to show enough conclusions. We selected clouds that occurred during the day and night in July and August 2017. We added a total of 60 cloud cases, 30 per month. Specifically, Table 1 shows information about cloud data. It's a good idea to add 2018 data. However, we didn't have enough time to get the 2018 data. Instead, we used data for two months in 2017 which are July and August, when tropical clouds were frequently observed each year. "

P 10

**Table 1. The number of selected convective clouds during observation time.**

| Month | Observation time | Sample number |
|---|---|---|
| July 2017 | 03:00-06:50 UTC (daytime) | 20 |
| | 21:00-24:50 UTC (nighttime) | 10 |
| August 2017 | 03:00-06:50 UTC (daytime) | 20 |
| | 21:00-24:50 UTC (nighttime) | 10 |

• Fig 1 & 2 only include the data based on the new satellite observations. I believe its worthwhile to include in the figures as well the behavior of the virtual (lower resolution) data. In this way it is easy for the reader to see the difference between the two.

"We totally agree with this comment. Figures have revised for the reader to compare high-resolution and low-resolution data. Figure 3 and 4 include high- and low-resolution data."

[Figure]

**Figure 3.** An example of temporal changes in minimum BT11 among thunderstorm pixels (10 August 2017, 03:10–05:50 UTC) for the high-resolution (2 km and 10 minutes) imagery (circle) and the low-resolution (4 km and 30 minutes) imagery (triangle). In this study, the lead time was defined as the time between the initial state and the mature state (time 0). The negative sign of time indicates the time ahead of the mature state of a tropical thunderstorm.

[Figure]

**Figure 4. An example of the thunderstorm development process through BT11 images (10 August 2017, 03:00–05:50 UTC); (a) the low-resolution (4 km and 30 minutes) imagery and (b) the high-resolution (2 km and 10 minutes) imagery.**

**Specific comments (chronological in appearance):**

**1. Introduction:**

• p1, L27, typo: "… lead to extensive economic losses"

"These severe events lead to extensive economic losses, environmental degradation, and subsequently, damage to human life."

• p2, L10/11: please rephrase "should be grounded" in this sentence. It is not clear what you mean.

"Removed"

• p2, L16: "(briefly, min)": what do you mean?

"Removed"

• p2, L20/21: "Moreover, … measurements": there are of course advantages using satellite in stead of ground-based observations, but also disadvantages. A network of radars, such as NEXRAD in the US or OPERA in Europe do provide high-resolution precipitation observations over a large area continuously with a higher spatial

resolution compared to satellite observations.

"This is very helpful comment. We added to the manuscript that some countries are using high-resolution radar system as well as satellites as follows."

P2 line 12-15

Aimed at disaster risk reduction, the European Operational Program for Exchange of Weather Radar Information (OPERA) continuously provides precipitation data of higher spatial resolution over a large area (Huuskonen et al., 2014). The Next Generation Weather Radar (NEXRAD) system (Klazura and Imy, 1993) has been employed for this purpose in the United States.

**2. Data and Method**

• p2, L30-p3,L2: so is this the only reason why the authors chose this particular region to investigate?

"The region examined in this study is from 10°N to 20°N and from 100°E to 120°E and is closely monitored by the Mekong River Commission. The Mekong River Commission is the only inter-governmental organization interacting directly with the governments of Cambodia, Lao PDR, Thailand, and Viet Nam to jointly manage the shared water resources and the sustainable development of the Mekong River (Jacobs, 2002). Unfortunately, this area is known as a vulnerable-disaster region because of its high risk of extreme weather. The global impacts of climate change have contributed to changes in the weather patterns that are felt across the Mekong River Commission region. The warmer atmosphere has the potential to contain more moisture, which increases the possibilities for thunderstorms invigoration under equal other conditions. The temperature increase associated with global climate change was generally assumed will to lead to increased thunderstorm intensity and associated heavy precipitation events (Schefczyk et al., 2015)."

[Figure]

**Figure 1. Himawari-8 AHI RGB image taken for this study area on 19 August 2015, 05:50 UTC. Several convective clouds (white color) are visible in the southern part of the area.**

• p3, L6: please rephrase " ...dramatically uprising in the clean sky"

"clouds which are developing within 2 hours in the clear sky"

• p3, L11,12: Regions 1, 2 and 3 are mentioned in the text but its not totally clear where those are located. A new figure indicating the three regions would clarify this.

"The observation range and time interval vary for each area in Himawari-8 AHI observation. In Southeast Asia, it does not belong to specific targeted regions which are from Region 1 and Region 5 like below.

[Figure]

**Subfigure 1. Observation time interval by targeted areas of Himawari-8 AHI.**

Thus, this study only observed Southeast Asia, high-resolution data provided per 10-minutes were used. Different time cycles of observation for each region are not covered in detail."

• p3, L14: Reference to JMA/MSC is now "2017", however, in the reference list it is "2018"

"Corrected" (JMA/MSC: Himawari-8/9 Imager (AHI), 2018)

• p3, L29: "… whose resolution is similar to the MTSAT ..."

"Corrected"

**3. Determining thunderstorm pixels and defining the lead time**

• p3, L31: please add a reference(s) for " … BT11 of clouds, which insofar, has been shown to be highly associated with the predictability of thunderstorms [references]"

"References are written and rearrange the paragraph"

P4 line 23-28

Sixty thunderstorms were subjectively selected based on the RGB images over Southeast Asia. The size of the selected thunderstorms was determined to be less than 120 km because such convective scales typically accompany precipitation (Houze Jr, 2004). We set the rectangular target boundaries depending on the thunderstorm size. In the target boundary, the BT11 values were monitored to determine the thunderstorm pixels and phases (initial/mature states) for the whole life cycle of thunderstorms. Since temporal changes in BT11 inform vertical drift velocity, the current status of clouds can be a key   diagnose the probability of imminent heavy rains/lightning soon (Vila et al., 2008).

• p4, L8: please rephrase "… time passed from when the ..." into " time in between the cloud … and ..."

[revised manuscript text omitted]

• p5, L18-L31: it would be worthwhile to include in Figure 2 the behaviour of the "virtual" lower resolution MTSAT data. In that way, the reader can check visually directly the difference for this particular case.

"That's a good point."

[Figure]

**Figure 3. An example of temporal changes in minimum BT11 among thunderstorm pixels (10 August 2017, 03:10–05:50 UTC) for the high-resolution (2 km and 10 minutes) imagery (circle) and the low-resolution (4 km and 30 minutes) imagery (triangle). In this study, the lead time was defined as the time between the initial state and the mature state (time 0). The negative sign of time indicates the time ahead of the mature state of a tropical thunderstorm.**

**5. Conclusions and limitation**

• p6, 1 sentence: this is expected. Even without this study one expects that newer instruments provide higher quality data, which in turn have a positive effect for any meteorological purpose. Exactly what you wrote on p6, L13-15.

"Changed"

P6 line5-11

In this paper, we compared an infrared channel at 10.45 µm of the high-resolution imagery and the low-resolution imagery. It was difficult to track the rapid BT11 changes in the clouds during the process of their development. The lead time of 60 cloud samples determined by the low-resolution imagery was not sufficiently accurately measured to monitor the whole development process of tropical thunderstorms. In contrast, the lead time of the high-resolution imagery was from 90 minutes and to 180 minutes before the cloud reached its mature state. Therefore, the higher spatial and temporal resolution of satellite observations can be valuable as it would alarm for tropical thunderstorms over Southeast Asia approximately two hours earlier than the low-resolution one based on the validation using 60 thunderstorms events.

• p6, L9-10: please rephrase " … and the mature deep convective with heavy rain"

"Corrected"

P6 line14-19

Second, future studies are needed to determine whether thunderstorms are rainy after the lead time. To more accurately examine the lead time, validation with surface precipitation data based on ground observation is further required. Also, the near-real radar data can be useful to validate the precipitation if the surface precipitation data have not been well managed. Finally, the lead time can differ depending on the region, since the lead time can be affected by various environmental factors such as wind direction and speed, atmospheric profiles, and the characteristics of the geolocation.

• p6, L23: "if applied to real technology". Do you mean " if implemented operationally?"

"Changed" Implemented operationally under real-life conditions, the high-resolution (2 km and 10 minutes) imager is of great significance to the provision of practical assistance to effective disaster management.

• p6, L26-27: " For example, Cambodia … satellite data are 4 km.": I am wondering if this is really the best place/appropriate (in a scientific article) referring to a specific country. I believe it is better to write in general terms that there are countries in southeast Asia who receive 4km data. For example, last line of the summary is written in more general words, which is in my opinion better.

"Absolutely, you are right. Thanks for your sincere comment!"

P6 line29-32

Although many developing countries in Southeast Asia use Himawari-8 satellite data, few countries carefully consider the importance of satellite resolution. A longer lead time is beneficial to the reduction of the risk of natural disasters caused by thunderstorms and the timely evacuation in such adverse events. Moreover, the low-resolution imagery is more effective in data storage than the low-resolution imagery.

**Tables:**

I think Table 1 & 3 could be transformed into 1 single table. The observation times in Table 1 can be put into Table 3. However, since more data will be included in the paper, the authors should think about how to restructure those tables. I can imagine that when you have not 8, but for example 80 observed clouds it would be better not to use a table but to make a figure of the distribution of the lead times & cloud scales … Is there maybe a relation between the cloud scale and the lead time? More things can be done when including more data!

"60 cloud examples are difficult to represent in a single table. The results are divided into Tables 1 and 3."

At the same time, thank you for suggesting new ideas related to this study. To analyze the correlation between cloud size and lead time, a preliminary test was conducted with 60 cloud samples. As a result, it was difficult to see a notable correlation between cloud scale and lead time. We consider that the number of samples will be increased to identify the relationship between cloud size and lead time. But, your idea is worth studying in the future. More detailed assumptions and definition of cloud-scale will be needed to precede the research.

(a) Correlation : 0.3755376

(b) Correlation : 0.3755376

[Figure]

**Subfigure 2. The correlation between the lead time and cloud scale; (a) the high-resolution (2 km and 10 minutes) imagery and (b) the low-resolution (4 km and 30 minutes) imagery.**

**Figures:**

• Figure 1: Please rephrase last sentence in the caption of this figure. At the moment, it is not clear. + Include the low-resolution data in this figure as well for direct comparison.

• Figure 2: I would like to see for Fig. 2 a & b that the authors include the virtual (lower resolution) data in order for the reader to see directly the difference between the new and old observational data.

**"All figures are updated for better understanding"**

[Figure]

**Figure 1. Himawari-8 AHI RGB image taken for this study area on 19 August 2015, 05:50 UTC. Several convective clouds (white color) are visible in the southern part of the area.**

[Figure]

**Figure 2.** Illustrations of 12 $\times$ 12-km pixels with different resolutions. The dark grey indicates the cloudy pixel, and the light grey indicates the clear-sky pixel. Only 2 cloudy pixels can be detected with the 4-km resolution imagery; in contrast, 18 cloudy pixels can be detected with the 2-km resolution imagery. The number of pixels at the cloud boundary varies depending on the resolution (Walker et al., 2012).

[Figure]

**Figure 3.** An example of temporal changes in minimum BT11 among thunderstorm pixels (10 August 2017, 03:10–05:50 UTC) for the high-resolution (2 km and 10 minutes) imagery (circle) and the low-resolution (4 km and 30 minutes) imagery (triangle). In this study, the lead time was defined as the time between the initial state and the mature state (time 0). The negative sign of time indicates the time ahead of the mature state of a tropical thunderstorm.

[Figure]

**Figure 4. An example of the thunderstorm development process through BT11 images (10 August 2017, 03:00–05:50 UTC); (a) the low-resolution (4 km and 30 minutes) imagery and (b) the high-resolution (2 km and 10 minutes) imagery.**

---

## Referee Report (RR1)

Manuscript nhess-2018-357 by Lee et al. investigates the effect of increasing resolution of geostationary satellite observations to predict tropical thunderstorms in Southeast Asia. The manuscript fits well within the scope of the NHESS journal.

The authors have taken into account many of the points raised related to the first version of the manuscript. However, some parts of the text should still be rewritten to increase readability:

Abstract:

L11: '…and a strong research interest exists in and efforts'

L14: 'time point'

Data and Method:

P3, L14: '… invigoration under equal other conditions'

P3, L15: '…was generally assumed will to lead …'

P3, L24-25: rephrase totally

P4, L11: '…in the area of 12 km.'

P4, L11: '…, whereas clouds and the light grey pixels'

P4, L18-19: '…establish the high-resolution imagery low-resolution imagery…'

Conclusion and limitation:

P6, L12-13: used 2 times 'this method may not work' in the same sentence

Summary:

P7, L3-L5: 'While the currently … is questionable': rephrase the sentence totally

---

## Author Response (AR2)

**Authors response-point by point**

[Final version] Answers to the comments of nhess-2018-357-referee-report-2.

We really thank to reviewer sincere and thoughtful comments. We tried to rephrase some parts of the text for improved readability. Thanks to the comments, the manuscript has been revised as follows before publication.

Abstract:

L11: '…and a strong research interest exists in and efforts'

"Tropical thunderstorms cause significant damage to property and lives, and a strong research interest exists in the advance and improvement of the thunderstorm predictability by satellite observations."

L14: 'time point'

"we examined the earliest possible time for prediction of thunderstorms as compared to the potential of low-resolution (4 km and 30 minutes) imageries of the former satellite."

Data and Method:

P3, L14: '… invigoration under equal other conditions'

"The warmer atmosphere has the potential to contain more moisture, which increases the possibilities for invigorating thunderstorms if other conditions are equal."

P3, L15: '…was generally assumed will to lead …'

"The temperature increase associated with global climate change was generally assumed to lead to increased thunderstorm intensity and associated heavy precipitation events (Schefczyk et al., 2015)."

P3, L24-25: rephrase totally

"Sixty clouds, except for those the already developed, were only selected the ones that whose cloudy pixels first began to detect at 03:00–06:50 UTC (daytime) and 21:00–24:50 UTC (nighttime) during in July and August 2017."

P4, L11: '…in the area of 12 km.'

"A tropical thunderstorm was found to be located in the area of $12 \times 12$-km pixels."

P4, L11: '..., whereas clouds and the light grey pixels'

"The dark grey indicates the detected cloudy pixel, and the light grey indicates the clear-sky pixel."

P4, L18-19: '…establish the high-resolution imagery low-resolution imagery…'

"Our final study aim was to quantitatively compare their effectiveness in the advanced predictability of tropical thunderstorms through the imageries of geostationary satellite."

Conclusion and limitation:

P6, L12-13: used 2 times 'this method may not work' in the same sentence

"First, this method may not work if a thunderstorm happens in the mesoscale convective system. In the mesoscale convective system, an intrusion of other developing clouds could frequently occur. Second, future studies are needed to determine whether thunderstorms are rainy after the lead time."

Summary:

P7, L3-L5: 'While the currently … is questionable': rephrase the sentence totally

"The currently used satellites can observe the area more precisely and their application to pre-disaster management are highly appreciated and demanded. This study examined the advance in the predictability of thunderstorms using geostationary satellite imageries."